# Triboelectric-Electromagnetic Hybrid Wind-Energy Harvester with a Low Startup Wind Speed in Urban Self-Powered Sensing

**DOI:** 10.3390/mi14020298

**Published:** 2023-01-23

**Authors:** Gang Li, Juan Cui, Tingshan Liu, Yongqiu Zheng, Congcong Hao, Xiaojian Hao, Chenyang Xue

**Affiliations:** Key Laboratory of Instrumentation Science & Dynamic Measurement, School of Instrument and Electronics, North University of China, Taiyuan 030051, China

**Keywords:** hybrid nanogenerators, triboelectric nanogenerator, electromagnetic generator, wind-energy harvesting, low startup wind speed, self-powered sensor

## Abstract

Wind energy as a renewable energy source is easily available and widely distributed in cities. However, current wind-energy harvesters are inadequate at capturing energy from low-speed winds in urban areas, thereby limiting their application in distributed self-powered sensor networks. A triboelectric–electromagnetic hybrid harvester with a low startup wind speed (LSWS-TEH) is proposed that also provides output power within a wide range of wind speeds. An engineering-implementable propeller design method is developed to reduce the startup wind speed of the harvester. A mechanical analysis of the aerodynamics of the rotating propeller is performed, and optimal propeller parameter settings are found that greatly improved its aerodynamic torque. By combining the high-voltage output of the triboelectric nanogenerator under low-speed winds with the high-power output of the electromagnetic generator under high-speed winds, the harvester can maintain direct current output over a wide wind-speed range after rectification. Experiments show that the harvester activates at wind speeds as low as 1.2 m/s, powers a sensor with multiple integrated components in 1.7 m/s wind speeds, and drives a Bluetooth temperature and humidity sensor in 2.7 m/s wind speeds. The proposed small, effective, inexpensive hybrid energy harvester provides a promising way for self-powered requirements in smart city settings.

## 1. Introduction

In recent years, the smart city concept has emerged as a solution to address challenges arising from the exponential growth of urban areas and population [1,2]. A key component of the concept is a network composed of distributed wireless sensors [3,4]. Energy efficiencies of such networks are critical factors in establishing the smart city [5]. However, current sensor networks rely mostly on batteries and cables for power supply at huge costs associated with installation and maintenance, drawbacks that potentially have environmental concerns [6]. To achieve continuous wireless power for the network, harvesting clean and renewable energy from the environment is desirable [7]. Wind energy, light energy, mechanical energy, and other environmental energy sources widely exist in cities [8,9]. Among them, wind energy has many advantages, such as its wide distribution in all-weather situations and ease of harvesting [10,11]. Average wind speeds in cities are generally 1.2 m/s to 3.8 m/s in Chinese cities from the National Centers for Environmental Information (NCEI). Such speeds are too low to start most wind-driven micro-harvesters employing electromagnetic generators (EMGs), and rectification loss is very high [12,13]. Therefore, the availability of energy harvesters able to operate in light to moderate breezes is an imperative of the smart city concept.

In 2012, the first triboelectric nanogenerator (TENG) proposed by Wang’s team showed great prospects in applications of self-powered systems [14]. With deeper research over the interim, TENGs have offered unique advantages in low-frequency energy collection, specifically, high output voltages from small input excitations [15,16,17]. They are widely used to harvest energy from environment, such as mechanical energy [18,19,20], tidal energy [21,22,23], and wind energy from various environments [24,25,26,27,28]. Currently, wind-energy harvesters operating with TENGs in the low-frequency regime have attained extensive attention because of their low cost, light weight, and high efficiency [29,30,31]. In cities, wind is characterized by low and broad average wind speeds. Moreover, natural wind patterns are unstable and highly variable. However, current TENGs and EMGs are incapable of achieving high energy conversion efficiencies under both low-speed and high-speed winds [32,33,34]. For better operations in urban settings, hybrid wind-energy harvesters combining both TENGs and EMGs may be the solution.

Recently, several prototypes of triboelectric–electromagnetic hybrid generators have been developed to harvest wind energy. Fan’s group proposed a self-powered wireless transmission sensor for the Internet of things, which uses the synchronous hybrid power generation mode of a TENG and an EMG to power the sensor [35]. Li’s group proposed an optimization strategy for flexible collaborative wind energy collection employing asynchronous operation of a TENG and an EMG; the strategy reduces the startup torque and the startup wind speed [36]. Among them, the startup wind speed of these hybrid energy harvesters is 2.2 m/s to 4 m/s [35,36,37,38,39]. However, to meet the energy supply demands of Bluetooth sensors, wind speeds of 4.7 m/s to 9 m/s are required to drive these harvesters [35,36,39]. Realizing the self-power supply needs of the wireless sensor network is difficult in breezes. Because the electromagnetic torque of the EMG has not been reduced, startup wind speeds of the energy harvesters are higher than the wind speeds for breezes, also limiting the output power increase of the TENG. Therefore, EMG designs for hybrid wind-energy harvester must be optimized to reduce the startup wind speed and increase the rotational speed.

With this objective, a triboelectric–electromagnetic hybrid harvester with a low startup wind speed (LSWS-TEH) is fabricated. Developed from the application of fluid mechanics theories in the design of the propeller, this harvester is designed with a low startup wind speed and high-power output over a wide range of wind speeds. In addition, an optimal design structure enables the inertia and electromagnetic torques of the EMG to be reduced, the speed of the harvester to be changed easily, and the resistance torque to be smaller. The harvester can operate and harvest wind energy in urban breezes with greater effectiveness. Experimental results show that its startup wind speed is 1.2 m/s; in wind speeds of 1.7 m/s, it can supply stable power to sensors with multiple integrated components. In addition, under wind speeds of 2.7 m/s, its output energy can steadily power a Bluetooth temperature and humidity sensor. The LSWS-TEH provides a benchmark for effective wind energy collection in breezes and, therefore, broad prospects in the field of the Internet of things connectivity and smart-city power planning.

## 2. Principles and Methods

### 2.1. Theoretical Analysis

#### 2.1.1. Design Principle for a Low Startup Wind Speed

During the startup of a wind harvester under low wind speeds, the mechanical model under steady rotor operations is mathematically expressed as
(1)$$\vec{M_{a}}-\vec{M_{f}}-\vec{M_{e}}-\vec{M_{m}}=\vec{M_{1}}+\vec{M_{2}}+\vec{M_{3}}\;,$$
where vectors $\vec{M_{a}}$, $\vec{M_{f}}$, $\vec{M_{e}}$, and $\vec{M_{m}}$ are the aerodynamic torque generated by the propeller, the frictional torque on the rotor, the eccentric torque on the propeller, and the electromagnetic torque applied to the electromagnetic rotor from Ampère forces, respectively. Vectors $\vec{M_{1}}$, $\vec{M_{2}}$, and $\vec{M_{3}}$ are the torques required for the rotation of the propeller, the electromagnetic rotor, and the other parts.

From Equation (1), the main measures to reduce the startup wind speed of the harvester are to reduce the frictional, eccentric, and electromagnetic torques, and to increase the aerodynamic torque of the propeller. Friction and eccentric torques are determined in the machining process and cannot be completely eliminated. Any reduction in electromagnetic torque affects the power-generation capacity of the electromagnetic generator. Therefore, to attain the objective, increasing the aerodynamic torque of the propeller is adopted.

Expanding the right-hand side of Equation (1),
(2)$$\vec{M_{1}}+\vec{M_{2}}+\vec{M_{3}}=\sum_{i}m_{i}r_{i}\vec{\omega }+\sum_{j}m_{j}r_{j}\vec{\omega }+\sum_{k}m_{k}r_{k}\vec{\omega },$$
where $\vec{\omega }$ is the angular velocity of the rotating part; the pairings ($m_{a}$, $r_{a}$) with a=i,j,k refer to the weight and radius of a differential element of the propeller, electromagnetic rotor, and other rotating parts, respectively.

From Equation (2), reducing the weight of the rotating part and decreasing the radius of rotation, while meeting structural strength requirements, can also effectively reduce the startup wind speed of harvesters. The design also exploits these two measures.

#### 2.1.2. Propeller Design

Considering the limitations on size, material, and processing technology, a flat blade is selected for the propeller for its ease in design and manufacture. According to fluid dynamics, the relationship linking inflow angle φ, attack angle α, and pitch angle β  is as follows:(3)φ=α+β .

The moving fluid produces a lift force that drives the propeller’s rotation and offers a more superior startup performance with low-speed winds. A high lift–drag ratio ξ is pursued to promote propeller rotation; ξ is calculated using
(4)$$\xi =\frac{C_{FBL}}{C_{FBD}}=\frac{\sin2\alpha }{2\sin^{2}\alpha },$$
where $C_{FBL}$ and $C_{FBD}$ are the lift and drag coefficients of the flat blade. The functional relationship between *ξ* and *α* shows that, to improve *ξ* in Figure 1a, *α* should be as small as possible.

Axial and tangential induction factors, *m* and *n*, are present during stable operations of the propeller; they are expressed as
(5)$$m=\frac{1}{3},$$
(6)$$n=\frac{m\;(1-m)}{\lambda^{2}}=\frac{2}{9\lambda^{2}},$$
where λ is the tip speed ratio. 

The inflow angle φ is calculated from
(7)$$\tan\varphi =\frac{1-m}{(1+n)\lambda }=\frac{6\lambda }{9\lambda^{2}+2}=\frac{6\lambda^{\prime }x}{9\lambda^{\prime }{}^{2}x^{2}+2},$$
(8)$$x=\frac{r}{R},$$
where $\lambda^{\prime }$ is the tip speed ratio of any blade element, *r* and *R* are the radii from the center of rotation to any blade element and the propeller edge, respectively, and *x* is the ratio of *r* to *R*, which is used to indicate the position of any blade element relative to the propeller.

Combining Equations (3) and (7) yields
(9)$$\varphi =\arctan\frac{6\lambda^{\prime }x}{9\lambda^{\prime }{}^{2}x^{2}+2}-\beta .$$

Although the blade elements of the propeller should have similar *α* for different *r*, the manufacturability of a propeller also needs to be considered. The pitch angle of the propeller is designed as
(10)$$\beta =\beta_{2}-(\beta_{2}-\beta_{1})\;x\;,$$
where $\beta_{1}$ and $\beta_{2}$ are the pitch angles at the tip and root of the blade, respectively.

With increasing wind speeds, the rotor speed is positively correlated with λ. For propellers of a low-speed wind turbine, the power coefficient increases to a peak and then decreases, whereas the torque coefficient monotonically decreases with λ in the range of 0–1.9 [40]. With λ between 0.6 and 1.2, the propeller obtains a sufficient torque while offering high efficiency in energy conversion. In comparing plots with λ equal to 0.6, 0.9, and 1.2 (Figure 1b), the curves of Equations (7) and (10) should generally be parallel in the range of *x* from 0.6 to 1 when λ is 0.9, thus reducing differences in attack angle α at different blade elements of the propeller. Moreover, as λ increases from 0.6 to 1.2, a family of curves generate from Equation (7) should be higher than the plots of Equation (10). The distance between the lowest point of family of curves and the point from Equation (10) is as small as possible when *x* is determined. Plotting the propeller parameter design curves (Figure 1b) shows that values of *x* between 0 and 0.4 easily produce stalling, which is not conducive to propeller rotation. Hence, positions of the propeller with *x* from 0 to 0.4 are simplified as belonging to the pillar.

The relationship between energy conversion efficiency and blade number can be found from Prandtl’s relation,
(11)$$\eta_{b}={\left(1-\frac{0.93}{b\sqrt{\lambda_{0}{}^{2}+0.445}}\right)}^{2}\;,$$
where *b* is the blade number, and $\lambda_{0}$ is the given tip speed ratio.

The first and second derivative functions of Equation (11) (Figure 1c) show that the energy conversion efficiency increases slowly with increasing blade number for certain $\lambda_{0}$. Considering that even-numbered blades are prone to resonance and engineering processing difficulties, a seven-bladed configuration is chosen.

The differential method is used to calculate the aerodynamic torque of the propeller,
(12)$$d\vec{M_{a}}=m\;\Delta \vec{\omega \;}r^{2}.$$

The blade should have large chord lengths at large radii, and high rotor solidity contributes to increasing the propeller aerodynamic torque.

In summary, a set of engineering solutions is found that established the main parameter settings of the propeller for the final design in Table 1.

### 2.2. Design and Working Principle of the Wind-Energy Harvester

The LSWS-TEH consists of a rotor, a stator, and a power management circuit (Figure 2). The rotor comprises a propeller, shaft, bearings, and a magnet. The stator includes housing, a coil, and coil holder. Figure 2a shows the rotor and stator assembly. The power management circuit is composed of a charge pump, rectifier bridge, capacitor bank, and switching regulator circuit (Figure 2b). The circuit schematic of rectification and charging circuit is shown in Figure 2c. The coil is wound onto the coil holder and is fixed to the stator housing (Figure 2d). The designed propeller and shell are made by 3D printing technology of SLA light-curing molding, and the material is white resin. The propeller has an overall size of 80 mm in diameter and 30 mm in cylindrical height; therefore, it is much smaller than other triboelectric–electromagnetic hybrid wind-energy harvesters [35,36,37,38,39].

When the ambient wind rotates the propeller, the polytetrafluoroethylene (PTFE) film in the TENG through centrifugation and rotation touches and slides against the aluminum foil. The output induced is an alternating current (AC) that flows between adjacent foils. A rectifier bridge converts AC to direct current (DC), which is then used to charge the capacitor bank. In the EMG, the propeller rotates magnets attached at the bottom of the shaft to produce a rotating electromagnetic field. The coil in the EMG cuts the rotating magnetic field lines to induce an electric potential. The output AC voltage is rectified and boosted by the charge pump to charge the capacitor bank. The switching voltage regulator then converts the unstable voltage of the capacitor bank into a stable voltage, which is a basic requirement of any sensor. If the power of the sensor is lower than the charging power of the capacitor bank, the excess power is stored in the capacitor bank to power the sensor when wind speeds are low.

Under different wind speeds, the TENG and EMG exhibit good synergy. In low-speed winds, the EMG output voltage is smaller than the forward bias voltage of the diode because the rotor rotates slowly, resulting in no voltage output after rectification. However, the TENG has high-voltage output characteristics that can generate a substantial voltage output under low-speed winds. Under high-speed winds, the rotor speed increases and the EMG generates a rectified voltage output. As speeds increase further, the output power and frequency from the TENG also increase.

The power generation principle of the TENG (Figure 3a) is detailed as follows: initially, the PTFE film and the aluminum foil are in contact with each other by centrifugation (Figure 3(aI)) to induce negative and positive electrostatic charges through contact friction. The PTFE film and the aluminum foil are gradually separated under rotation, and the potential difference between adjacent electrodes generates an AC through the external circuit (Figure 3(aII)). The PTFE then makes complete contact with the next electrode (Figure 3(aIII)), meaning that one charge transfer is performed. Next, the adjacent electrodes produce an opposite potential difference to that previously, and the opposite current flows in the external circuit (Figure 3(aIV)). The above cycle is repeated with the TENG outputting a continuous AC. Using COMSOL Multiphysics 5.6 software, the electric potential distribution on the surface of the TENG (Figure 3b) is simulated at four stages during the cycle.

The working principle of the EMG is shown in Figure 3c. At the beginning, a zero rate of change in magnetic flux through the coil exists, and the coil provides no voltage output (Figure 3(cI)). At its maximum rate of change (Figure 3(cII)), the induced electric potential in the coil peaks and the current flowing through the external circuit reach a peak. Figure 3(cIII) is similar to Figure 3(cI), but the magnetic field distribution is reversed. Figure 3(cIV) is similar to Figure 3(cII), but the electric potential and current are the opposite of the previous ones. By repeating the cycle, an induced AC is generated through the EMG, and its surface flux density is also simulated using COMSOL software (Figure 3d), with the four stages exhibiting typical fields of the EMG during an AC cycle.

### 2.3. Experimental Methods

#### 2.3.1. Fabrication of the EMG

One copper wire with diameter of 0.1 mm is wound on the resin holder as two coils. The coils are positioned at a distance of 5 mm. Each coil has 400 turns, and the width and height of the coil are 12.5 mm and 11 mm, respectively. A neodymium (NdFeB) disc magnet with a radius of 4 mm and height of 3 mm is attached to shaft with a diameter of 1.5 mm.

#### 2.3.2. Fabrication of the TENG

Aluminum foil with a thickness of 0.06 mm is used as the metal electrodes. These electrodes are bonded to copper foil of thickness 0.06 mm. The rotating triboelectric electrode is selected as a PTFE film with a height of 20 mm and is glued to the paddle housing.

#### 2.3.3. Testing Systems

The harvester is fixed with a stand in a wind tunnel controlled by an axial flow fan (SFG3-2R, Leifeng Mechatronics Co., Changzhou, China) operating through a frequency converter (SQ1000−2T, Fuci Electromechanical Technology Co., Shanghai, China). The output voltage signal is obtained using an oscilloscope (DSOX3024T, Keysight, Santa Rosa, CA, USA) through a 100 M high-voltage probe (PA5100A, Tektronix, Beaverton, OR, USA), and the current signal is obtained using an electrostatic meter (6514, Keithley, Beaverton, OR, USA). The wind speed is obtained using a thermal anemometer (AR866A, Kexin Measurement & Control Technology Co., Suzhou, China).

The electrical outputs of EMG and TENG are independent. In the process of the output capability test and impedance test of the harvester, the outputs of EMG and TENG are tested separately; that is, only EMG or TENG is connected to the test system during the test. Thus, the measured data reflect the electrical outputs of the single EMG or TENG.

## 3. Results and Discussion

### 3.1. Output Performance

The angle between the chord of the propeller and the horizontal plane (pitch angle) was determined, and the effect of propeller curvature on the EMG output was investigated in four different scenarios (Figure 4). In this experiment, the output performance of the EMG was tested with propeller curvatures of 0° (plane), 50.6°, 24.7°, and 16.4°. With a curvature of 50.6°, the arc at the propeller tip was tangent to the horizontal plane; in this instance, the radius of the arc was denoted by $r^{\prime }$. Then, the curvature of the propeller was 24.7° at an arc radius of $2r^{\prime }$ and 16.4° at an arc radius of $3r^{\prime }$.

The EMG output decreased more significantly over all wind speeds when the curvature is large (Figure 4a,b). Curvature could improve airflow rate into the propeller, but larger curvatures created vortices that reduced the EMG output. The EMG performance at a wind speed of 1.2 m/s (Figure 4a,c) worsened when the curvature was in the midrange, whereas it showed greater improvements at wind speeds of 1.2 m/s and 1.5 m/s for small curvatures (Figure 4a,d). That is, a suitable curvature could improve the aerodynamic performance of the propeller under low-speed winds. For the application, a propeller curvature of 16.4° was chosen.

Next, the effect of rotating triboelectric electrode length on output performance of the TENG was studied using two metal electrodes in experiments in which the external circuit of the EMG was short-circuited. For the rotating triboelectric electrodes, strips of PTFE film with lengths of 3 cm, 4 cm, and 5 cm and thickness of 0.1 mm were tested (Figure 5). The 3 cm electrode had the lowest startup speed, and the startup wind speed increased with increasing strip length, as shown in Figure 5 (aI–cI). As shown in Figure 5 (aI–cI), the 4 cm long electrode had the highest output voltage because its stiffness produced a suitable curvature, resulting in the highest effective contact area for the TENG. As shown in Figure 5, the 5 cm electrode produced the lowest output power under low-speed winds, due to a reduced effective contact area because of its poor stiffness and easy deformation. It had the highest output current because its stronger centrifugal action increased the effective contact area under high-speed winds.

By measuring the frequency of the EMG or TENG output voltage, the rotation speed of the harvester can be found by
(13)$$n=60f_{EMG}=\frac{120f_{TENG}}{p},$$
where n is rotation speed, $f_{EMG}$ is the frequency of the EMG, $f_{TENG}$ is the frequency of the TENG, and p is the number of TENG metal electrodes.

In addition, calculating the ratio k, i.e., the ratio of the rotating triboelectric electrode length to the number of metal electrodes, the TENG output was found to rise and fall with k. Optimizing k could improve the TENG output performance. The 3 cm electrode had the lowest startup wind speed; hence, it was chosen to balance the startup wind speed with the electrical energy output of TENG.

Lastly, the effect of different numbers of metal electrodes on TENG output performance was investigated without changing the total length of the metal electrodes. The output conditions for the TENG with two metal electrodes are given in Figure 5a. With the same strip length of 3 cm, the TENG electrical outputs obtained with four, six, and eight metal electrodes (Figure 6) show that, with increasing number, the outputs displayed a peak, and the highest output power was achieved with six metal electrodes. Under the same wind speed, higher numbers of metal electrodes generated a higher energy conversion frequency. However, increasing the electrode number further decreased the charge accumulation time that affected the generation of large voltages. Therefore, the harvester design was configured with six metal electrodes.

The final design parameters for the harvester were determined according to the results of Section 2.1.2 and the above experiments and analysis.

### 3.2. Demonstration

The TENG hinders rotor rotation; hence, the changes in peak power and the rates of change in the short-circuit current of the EMG containing the TENG and of the EMG alone were compared for different wind speeds (Figure 7a). At wind speeds less than 3 m/s, the output power for the EMG of LSWS-TEH was slightly reduced under the same wind speeds. The output voltage of the EMG under low-speed winds was too small to reach the forward bias of the diode, resulting in no electrical output after rectification. In contrast, the LSWS-TEH had a higher electric energy output range compared with that for the EMG because an electric energy output was also present under low-speed winds, attributable to the high-voltage output of the TENG. Although the output power of the harvester decreased slightly in low-wind-speed conditions, it meets the requirements for urban applications in which the average wind speed ranges from 1.2 m/s to 3.8 m/s. Furthermore, high rotation speeds enhance centrifugal forces, which may destroy the balance within the wind-energy harvester and break the propeller. For self-protection, large wind turbines usually lock the propeller. The TENG can protect the harvester during power generation under high-speed winds because friction increases when centrifugal effects increase.

The peak output power of the TENG (Figure 7(bI)) and EMG (Figure 7(bII)) under different wind speeds and impedances was investigated. When the harvester was connected to an external load, it could be regarded as a circuit model where the internal resistance and external load divided the open-circuit voltage. Therefore, the output voltage and output current were positively and negatively correlated with the external load, respectively. Furthermore, as the load increased, the output power of TENG and EMG firstly increased and then decreased (Figure 7b). The corresponding optimum resistances for TENG and EMG were about 100 MΩ and 90 Ω, respectively. The charging of a 100 μF capacitor separately by the EMG, TENG, and LSWS-TEH under wind speeds of 2 m/s (Figure 7c) show that the harvester had a higher charging speed and higher capacitance voltage than either EMG or TENG. Moreover, with multiple integrated components, the harvester could power a sensor with a rated power of 150 μW (Figure 7d). In addition, under wind speeds of 2.7 m/s, it could power a Bluetooth temperature and humidity sensor (Figure 7e). With a rated power of 16 mW, the sensor could transfer data to a cell phone APP. Overall, the harvester delivered excellent performances in low-speed winds and had a wide wind speed collection range, meeting the energy-harvesting requirements in urban settings.

The aerodynamic torque, $M_{a}$, affects the starting wind speed of the wind-energy harvesters; the torque output of the propeller is
(14)$$M_{a}=\frac{1}{2}\;\rho C_{m}RSV^{2}=\frac{1}{2}\pi \;\rho C_{m}R^{3}V^{2},$$
where ρ is the air density, $C_{m}$ is the moment coefficient, *R* is the propeller radius, *S* is the cross-sectional area of the propeller rotating body, and *V* is the wind speed. The torque output is proportional to the cube of *R* and the square of *V*.

The power output, *P*, can be calculated by
(15)$$P=\frac{1}{2}\;\rho C_{p}SV^{3}=\frac{1}{2}\;\pi \rho C_{p}R^{2}V^{3},$$
where $C_{p}$ is the power factor. The power output is proportional to the square of R and the cube of V.

In general, obtaining a high power output is difficult at low wind speeds in small volumes. However, the designed LSWS-TEH achieves a low startup wind speed in a small volume and has a more substantial peak power output, which can drive the sensors at lower wind speed. The performances of different triboelectric–electromagnetic hybrid wind-energy harvesters were compared, as shown in the Appendix A. The proposed LSWS-TEH with a smaller size showed a lower startup wind speed and a higher output power.

## 4. Conclusions

The LSWS-TEH comprising a TENG and an EMG was developed to operate in 1.2 to 11 m/s wind speeds. It has an extremely low startup wind speed and wide operating range. An engineering design strategy was developed for the propeller, which substantially improved the aerodynamic torque of the propeller; accordingly, the harvester was able to meet the requirements for wind-energy harvesting in ultralow and medium–high wind speed in city settings. The TENG resolves the disadvantages inherent in the EMG without output power after rectification at low-speed winds and mitigates propeller damage under high-speed winds. Through specific design features of the structure, both inertial and electromagnetic torques of the EMG were reduced. Experiments demonstrated that the harvester started under wind speeds of 1.2 m/s. A sensor with multiple integrated components was powered using the harvester in wind speeds of 1.7 m/s. Moreover, a Bluetooth temperature and humidity sensor was operational under wind speeds of 2.7 m/s. The harvester can collect wind energy in ultralow and medium–high wind speeds and has wide potential in applications requiring self-powered wireless sensor networks operating in smart city environments.

## Figures and Tables

**Figure 1 micromachines-14-00298-f001:**
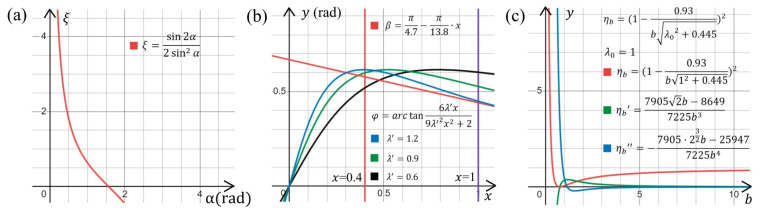
Relationship analysis for propeller parameter selection (see key legends for parameter expressions and settings): (**a**) between *ξ* and *α*; (**b**) between *φ*, *α*, and *β*; (**c**) between *b* and *η_b_*.

**Figure 2 micromachines-14-00298-f002:**
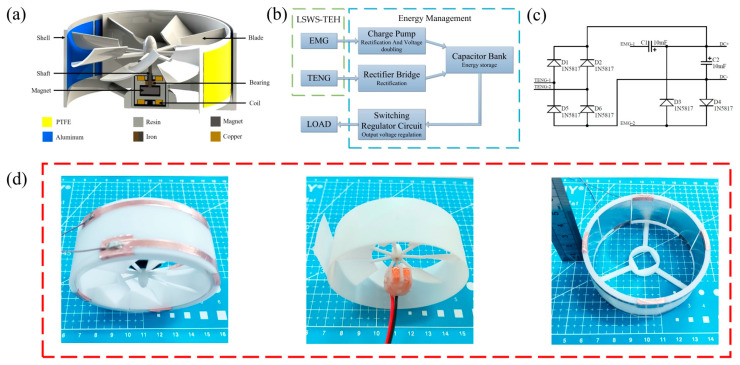
Design and photos of LSWS-TEH: (**a**) structure schematic of the rotor and stator; (**b**) schematic of circuit block diagram; (**c**) rectification and charging circuits; (**d**) three photos show the LSWS-TEH, coils and propeller, housing respectively.

**Figure 3 micromachines-14-00298-f003:**
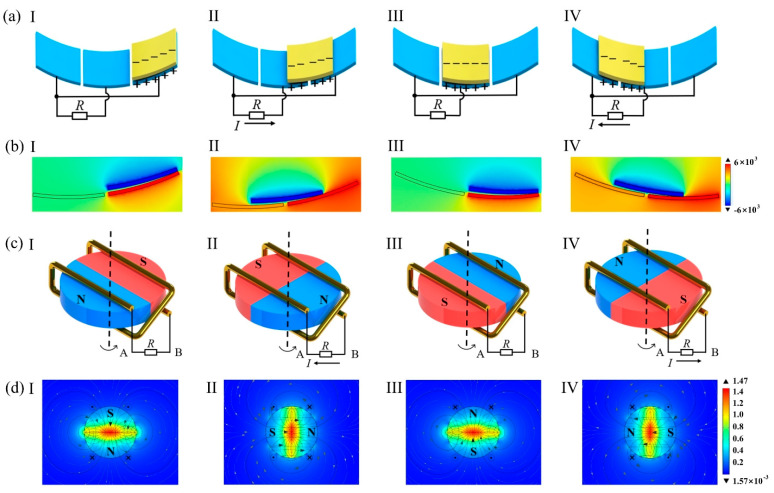
LSWS-TEH operating principle and simulation results: (**aI**) working schematic of the TENG state 1; (**aII**) working schematic of the TENG state 2; (**aIII**) working schematic of the TENG state 3; (**aIV**) working schematic of the TENG state 4; (**bI**) simulation of the surface electric potential distribution of the TENG state 1; (**bII**) simulation of the surface electric potential distribution of the TENG state 2; (**bIII**) simulation of the surface electric potential distribution of the TENG state 3; (**bIV**) simulation of the surface electric potential distribution of the TENG state 4; (**cI**) working schematic of the EMG state 1; (**cII**) working schematic of the EMG state 2; (**cIII**) working schematic of the EMG state 3; (**cIV**) working schematic of the EMG state 4; (**dI**) simulation of the surface flux density of the EMG state 1; (**dII**) simulation of the surface flux density of the EMG state 2; (**dIII**) simulation of the surface flux density of the EMG state 3; (**dIV**) simulation of the surface flux density of the EMG state 4.

**Figure 4 micromachines-14-00298-f004:**
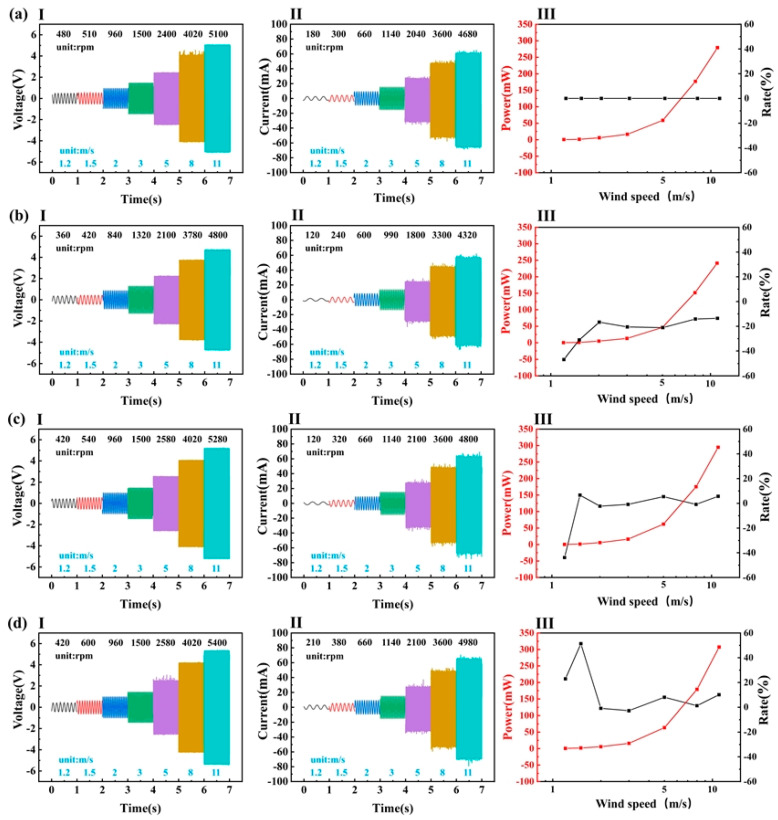
Open-circuit voltage, short-circuit current, power, and rate of change in power of the EMG for different blade curvatures: (**aI**) open-circuit voltage of EMG in flat; (**aII**) short-circuit current of EMG in flat; (**aIII**) power and rate of EMG in flat; (**bI**) open-circuit voltage of EMG in 50.6°; (**bII**) short-circuit current of EMG in 50.6°; (**bIII**) power and rate of EMG in 50.6°; (**cI**) open-circuit voltage of EMG in 24.7°; (**cII**) short-circuit current of EMG in 24.7°; (**cIII**) power and rate of EMG in 24.7°; (**dI**) open-circuit voltage of EMG in 16.4°; (**dII**) short-circuit current of EMG in 16.4°; (**dIII**) power and rate of EMG in 16.4°.

**Figure 5 micromachines-14-00298-f005:**
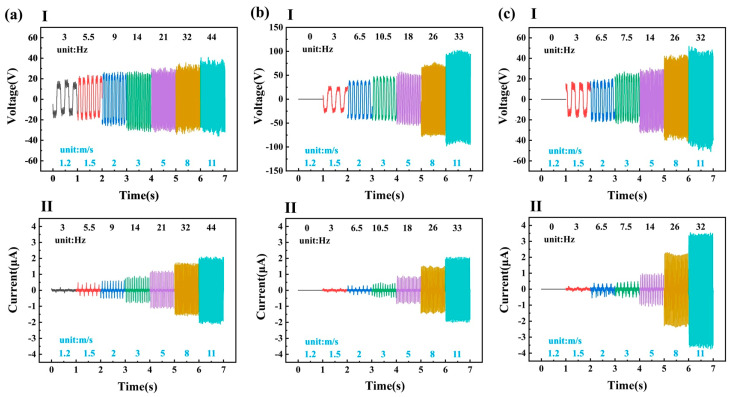
Open-circuit voltage and short-circuit current produced by the TENG for different strip lengths of PTFE film: (**aI**) open-circuit voltage of TENG in 3 cm film; (**aII**) short-circuit current of TENG in 3 cm film; (**bI**) open-circuit voltage of TENG in 4 cm film; (**bII**) short-circuit current of TENG in 4 cm film; (**cI**) open-circuit voltage of TENG in 5 cm film; (**cII**) short-circuit current of TENG in 5 cm film.

**Figure 6 micromachines-14-00298-f006:**
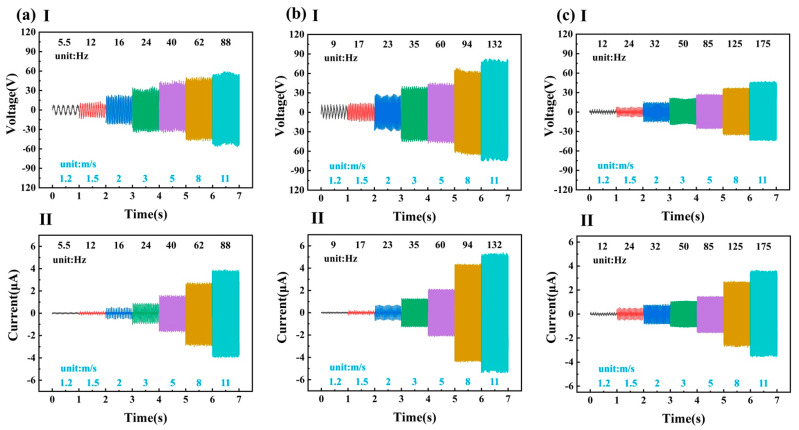
Open-circuit voltage and short-circuit current of the TENG for three different numbers of metal electrodes: (**aI**) open-circuit voltage of TENG in 4 electrodes; (**aII**) short-circuit current of TENG in 4 electrodes; (**bI**) open-circuit voltage of TENG in 6 electrodes; (**bII**) short-circuit current of TENG in 6 electrodes; (**cI**) open-circuit voltage of TENG in 8 electrodes; (**cII**) short-circuit current of TENG in 8 electrodes.

**Figure 7 micromachines-14-00298-f007:**
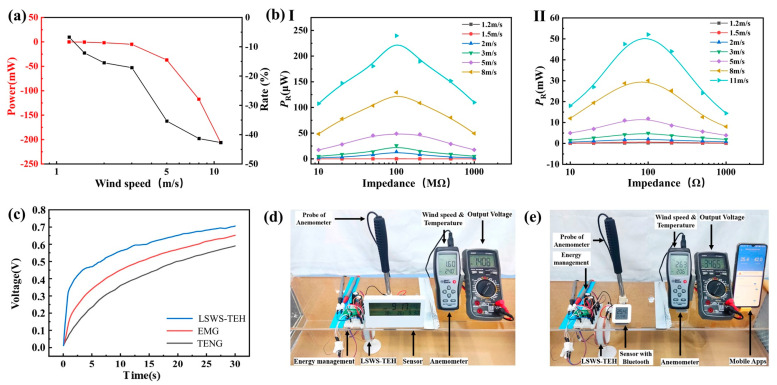
The LSWS-TEH performance demonstration under various wind speeds: (**a**) change in peak power and rate of change for the short-circuit current; (**bI**) peak power outputs for the TENG; (**bII**) peak power outputs for the EMG; (**c**) charging of a 100 μF capacitor with rectifier bridge under a 2 m/s wind speed; (**d**) powering of a sensor with multiple integrated components under wind speeds of 1.7 m/s; (**e**) powering of a Bluetooth temperature and humidity sensor under wind speeds of 2.7 m/s.

**Table 1 micromachines-14-00298-t001:** Settings of the main parameters of the propeller ^1^.

Designed Tip Speed Ratio	Blade Root Pitch Angle	Blade Tip Pitch Angle	Number of Blades	Rotor Solidity ^2^
0.6–1.2	38.3°	25.3°	7	90%

^1^ The propeller is simplified as the pillar with *x* in the range of 0 to 0.4. ^2^ The rotor solidity is calculated without simplified blades.

## Data Availability

The data presented in this study are available on request from the corresponding author.

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
