# Peer review of "Triboelectric-Electromagnetic Hybrid Wind-Energy Harvester with a Low Startup Wind Speed in Urban Self-Powered Sensing"

_micromachines, 2023, doi:10.3390/mi14020298_

Round 1

Reviewer 1 Report

The manuscript could be published after the following minor revision and a detailed explanation:

1.      The authors should compare their LSWS-TEH with other Triboelectric-electromagnetic hybrid wind energy harvesters on performances including start-up wind speed, output voltage, output current, operating range, service life and power generated.

2.      How to fabricate the designed propeller? 3D printing? What material?

3.      What is the matching impedance of the device?

Reviewer 2 Report

In this work, the authors developed a triboelectric-electromagnetic hybrid harvester with a low start-up wind speed to provide output power within a wide range of wind speeds. This manuscript is well-organized and can be improved after minor revision before publication. Some points should be considered as follows:

How could the author know which part of the voltage is generated by EMG/TENG?

How does the device’s voltage, current, and power change when the device is connected to external load of different resistances?

The charging circuit scheme should be incorporated in Figure 4c as an inset for the readers to understand the charging mechanism.

The relationship between wind speed and device rotation speed is suggested to be provided as the device rotation contributes to the voltage output directly.

Compared with other kinds of EMG/TENG-based wind harvesters, what are the advantages of LSWS-TEH developed in this study?
